# Effect of the Association of Fixed Oils from *Abelmoschus esculentus* (L.) Moench, *Euterpe oleracea* Martius, *Bixa orellana* Linné and Chronic SM^®^ on Atherogenic Dyslipidemia in Wistar Rats

**DOI:** 10.3390/molecules28186689

**Published:** 2023-09-19

**Authors:** Priscila Faimann Sales, Aline Lopes do Nascimento, Fernanda Cavalcante Pinheiro, Andressa Ketelem Meireles Alberto, Abrahão Victor Tavares de Lima Teixeira dos Santos, Helison de Oliveira Carvalho, Gisele Custódio de Souza, José Carlos Tavares Carvalho

**Affiliations:** 1Laboratory of Drugs Research, Biology and Healthy Sciences Department, Pharmacy Faculty, Federal University of Amapá, Rod. JK, Km 02, Amapá, Macapá 68902-280, Brazil; pfaimann@gmail.com (P.F.S.); ali.nascimento99@gmail.com (A.L.d.N.); fernandacavalcante602@gmail.com (F.C.P.); andressaketelem@gmail.com (A.K.M.A.); abrahaolima28@gmail.com (A.V.T.d.L.T.d.S.); helison.farma@gmail.com (H.d.O.C.); gi.custodio.souza@gmail.com (G.C.d.S.); 2University Hospital of Federal University of Amapá, Rodovia Josmar Chaves Pinto, Macapá 68903-419, Brazil

**Keywords:** hypercholesterolemia, atherosclerosis, vegetable oils

## Abstract

Dyslipidemia presents high levels of serum cholesterol and is characterized as a risk factor for cardiovascular diseases, especially for the development of atherosclerosis. *E. oleracea* oil (OFEO), *A. esculentus* oil (OFAE), *B. orellana* oil (OFBO), and Chronic SM^®^ granules (CHR) are rich in bioactive compounds with the potential to treat changes in lipid metabolism. This study investigated the effects of treatments with oils from *A. esculentus*, *E. oleracea*, *B. orellana*, and Chronic SM^®^ on *Cocos nucifera* L. saturated-fat-induced dyslipidemia. The chromatographic profile showed the majority presence of unsaturated fatty acids in the tested oils. The quantification of tocotrienols and geranylgeraniol in OFBO and CHR was obtained. Treatments with OFEO, OFAE, OFBO, and CHR were able to significantly reduce glycemia, as well as hypertriglyceridemia, total cholesterol, and LDL-cholesterol, besides increasing HDL-cholesterol. The treatments inhibited the formation of atheromatous plaques in the vascular endothelium of the treated rats. The obtained results suggest that the OFEO, OFAE, OFBO, and CHR exhibit antidyslipidemic effects and antiatherogenic activity.

## 1. Introduction

Dyslipidemia occurs due to changes in the plasma concentration of low-density lipoprotein (LDL), high-density lipoprotein (HDL), and triglycerides (TGs) [1]. Dyslipidemia is classified as isolated hypercholesterolemia, isolated hypertriglyceridemia, mixed hyperlipidemia, and low HDL-cholesterol [2]. Increased concentrations of LDL-cholesterol and triglycerides and decreased concentrations of HDL-cholesterol are major risk factors for cardiovascular disease [3].

Many pathological conditions can compromise the cardiovascular system, such as atherosclerosis, a chronic immunoinflammatory condition triggered mainly by modifications to the lipid metabolism and reactive oxygen species (ROS) [4]. Atherosclerosis is characterized by the hardening and progressive narrowing of blood vessels due to the interaction between arterial wall elements, modified lipoproteins, T cells, macrophages, and platelets, resulting in the appearance of atherosclerotic plaques, which hinder the supply of blood to the tissues [5].

Multiple epidemiological studies associate diet composition as the main risk factor for dyslipidemia [6]. Currently, there are several drugs capable of controlling hyperlipidemia, but they are expensive and not free of side effects. Thus, therapeutic alternatives of natural origin constitute promising sources of new functional bioactive substances as a treatment or adjunct to conventional treatments of chronic diseases, such as hyperlipidemias [7,8].

Thus, the general population has opted for a diet with functional compounds that have cardioprotective properties, such as unsaturated fatty acids [7]. Unsaturated fatty acids (UFAs) are obtained through the diet from vegetable oils and are classified as monounsaturated fatty acids (MUFAs), with only one double bond, and polyunsaturated fatty acids (PUFAs), containing two or more double bonds. The intake of unsaturated fatty acids can prevent and support the treatment of cardiovascular diseases, aiding, therefore, in the reduction in the biochemical parameters, decreasing the plasma concentrations of triglycerides through a decrease in the synthesis of triglycerides by the liver, as well as in an increase in the activity of lipoprotein lipase (LPL), accelerating the catabolism of very-low-density lipoprotein (VLDL) and chylomicrons [9].

Numerous plant species are producers of oilseeds, whose oils present in their composition bioactive properties of great potential for cosmetics, food, and pharmaceutical applications [10]. The vegetable oils from the seeds present essential fatty acids, free of trans fats or cholesterol and fat-soluble vitamins [11]; these fatty acids perform activities of great biological importance for human health, acting as regulators of several key genes in the modulation of anti-inflammatory, anti-hypercholesterolemic, and antioxidant responses, thus triggering cardioprotective effects [12].

Among these promising oilseed species is *Abelmoschus esculentus* (L.) Moench (okra), a shrub with greenish capsule fruits, containing spherical seeds with high mucilage concentrations. The seeds contain oil at a concentration of 20 to 40%, which can be equated to the composition of oilseeds such as soybeans, peanuts, and sesame [13,14]. The seeds are sources of vitamin E, a powerful natural antioxidant of great nutritional importance [15], and have phenolic compounds, flavonoids, catechins, and oligomeric derivatives; they are also sources of triacylglycerides, phytosterols, and phospholipids, and also have unsaturated fatty acids, mainly polyunsaturated fatty acids [16,17,18].

*Euterpe oleracea* Martius, known as the açaí tree, is a fruit tree of great socioeconomic relevance in the Amazon, with smooth globose fruits, violet in color, known as açaí. From the pulp of the fruit, a dark green oil is obtained with an aroma distinct to the açaí [19]. The açaí oil accounts for 50% of the total dry matter of the pulp and has phenolic compounds, mainly anthocyanins and dietary fiber, in addition to being a good source of minerals such as potassium, magnesium, calcium, phosphorus, sodium, and vitamins E and B1, and it also has a lipid profile rich in mono- and polyunsaturated fatty acids [20,21].

The species *Bixa orellana Linné*, known in Brazil as annatto, produces a capsular fruit containing reddish-orange seeds inside [22,23]. The oil extracted from the seeds of *B. orellana* is of great scientific interest due to its high amount of bioactive compounds. It presents in its composition a lipid fraction of mono- and polyunsaturated fatty acids of great biological importance [24]. The seed oil contains compounds such as alkaloids, flavonoids, and carotenoids, in addition to having a large amount of tocotrienols, mainly δ-tocotrienol, as well as high concentrations of geranylgeraniol [25,26].

The Chronic SM^®^ granulate is a nutraceutical developed and patented by the helthspan company Ages Bioactive Compounds, consisting of *E. oleracea*, *A. esculentus*, and *B. orellana* oils. It has high bioavailability guaranteed by the Evolve^®^ technology, which consists of a coated system that makes the oily compounds available in their solid form, ensuring the protection of unsaturated fatty acids, the preservation of phytoactives, and increased absorption.

Several studies have attributed promising activities to unsaturated fatty acids, acting in the control and prevention of diseases, such as in lipid metabolism disorders, especially in the development of dyslipidemia [27,28]. In this study, we aimed to investigate the possible effects of the treatments with oils from *A. esculentus*, *E. oleracea*, *B. orellana*, and Chronic SM^®^ granules on dyslipidemia induced by saturated fat *Cocos nucifera* L. in rats.

## 2. Results and Discussion

### 2.1. Chemical Composition of Euterpe oleracea Martius Oils; Abelmoschus esculentus L. Moench

The intake of lipids in the human diet triggers an increase in free fatty acids that can raise cholesterol levels, which contributes to the process of dyslipidemia and is one of the main factors in the occurrence of cardiovascular diseases. The high concentration of saturated fatty acids is among the main lipids that favor hyperlipidemia and raise serum levels of triglycerides (TG), total cholesterol (TC), and low-density lipoprotein cholesterol (LDL-c), which are responsible for the development of atherosclerosis and chronic inflammation in blood vessels [1,29].

The oils obtained from plant species present variations In their chemical composition according to the degree of unsaturation of each species and are rich in essential fatty acids free of trans fats, cholesterol, and fat-soluble vitamins. The chemical composition of vegetable oils in relation to the presence of fatty acids can determine the use of these oils for various purposes, such as food and/or therapy [20].

In the analysis by gas chromatography coupled to mass spectrometry (GC-MS), we can identify six retention peaks of compounds (Figure 1). According to the *Euterpe oleracea* fixed oil (OFEO) chromatographic data (Table 1), we observed a higher percentage of unsaturated fatty acids (67.83%), with polyunsaturated (5.9%) and monounsaturated (61.27%). Mostly, we can find in its composition oleic acid (54.32%), palmitic acid (30.0%), and linolenic acid (5.9%), followed by elaidic acid (4.29%), palmitoleic acid (2.62%), and stearic acid (2.29%). The data presented are in accordance with the literature and are present in 71% of unsaturated fatty acids, of which 60.81% are monounsaturated and 10.36% are polyunsaturated. Predominant in its composition are oleic acid (56.2%) and palmitic acid (24.1%) [19,30,31].

Through chromatographic analysis of the fatty acids present in *Abelmoschus esculentus* fixed oil (OFAE), it was possible to identify five compound retention peaks (Figure 2). We can observe a higher percentage of unsaturated fatty acids (61.05%), polyunsaturated (42.12%) and monounsaturated (19.38%) (Table 2). This result is in line with other authors, since large amounts of unsaturated lipids (66.32%), especially oleic (20.38%) and linoleic (44.48%) acids, were found in okra grown in northeastern Brazil [32]; variations of linolenic acid (23.6 to 50.6%) and palmitic acid (10.03 to 36.03%) were also observed [33]. However, we observed in this work the presence mostly of linoleic acid (42.12%) and palmitic acid (33.55%), followed by oleic acid (17.85%), stearic acid (4.96%), and gondoic acid (1.53%) in the OFAE, thus presenting a similarity with those described in the literature.

The calibration curve obtained for the total tocotrienols based on delta-tocotrienol presented R^2^ = 0.9904, obtaining the following straight line equation: Y = 5,000,000x – 857,622 (Figure 3). From the quantification of total tocotrienols, it was possible to obtain the average concentration of 617.44 mg/mL, with an average content of 61.74% for *Bixa orellana* fixed oil (OFBO), and for Chronic SM^®^ granules (CHR), it was possible to obtain 467.56 mg/mL, with an average content of 46, 75% of total tocotrienols (Table 3).

The calibration curve obtained for geranylgeraniol showed R2 = 0.9989, obtaining the following straight line equation: Y = 20,000,000x – 400,000 (Figure 4). From the quantification of geranylgeraniol present in the samples, it was possible to obtain an average concentration of 872.13 mg/mL for OFBO and an average content of 87.21% of geranylgeraniol; for CHR, it was possible to obtain 141.69 mg/mL and an average content of 14.16% of geranylgeraniol (Table 4).

### 2.2. Analysis of Clinical and Biochemical Parameters

Diets rich in saturated fatty acids (SFAs) increase insulin resistance and the incidence of cardiovascular disease. While those rich in mono and polyunsaturated fatty acids perform important roles in the adjuvant treatment of heart disease, coronary disease, hypertension, dyslipidemia, type 2 diabetes mellitus, and insulin resistance [33,34,35,36,37,38,39,40,41,42].

MUFAs such as PUFAs act in the modulation of the lipid profile, especially the fatty acids of the ω-3, ω-6, and ω-9 series that are present in the composition of oils; these fatty acids are regulators of the expression of genes involved in lipid metabolism and glucose and adipogenesis and may act via mediators, such as the peroxisome proliferator-activated receptor (PPAR) (α, β, and γ), hepatic X receptors (LXRs) (α and β), hepatic nuclear factor receptors (α and β), 4 (HNF4-α), and sterol regulatory element-binding proteins (SREBPs) 1 and 2, which are important factors in the hepatic metabolism of carbohydrates, fatty acids, triglycerides, cholesterol, and bile acids [43,44,45].

The activation of the PPARγ transcription factor is essential in adipogenesis. Certain fatty acids of the ω-3 and ω-6 series can act as PPARγ ligands and thus control adipogenesis by reducing cholesterol and triglyceride levels, in addition to contributing to the reduction in adipose tissue formation [46]. Thus, we can observe in our data a reduction in the body weight of the treatment groups CHR, OFEO, OFAE, and OFBO. The reduction in body weight may be associated with the high levels of AGIs present in these oils and in the granules. The CHR (259.57 ± 4.00) and OFBO (243.52 ± 4.25) groups, both with *p* < 0.001, had lower body weight and daily feed intake when compared to the *Cocos nucifera* L. saturated fat group (GSC) (Table 5).

The supplementation of UFA content is also associated with the reduction in hyperplasia and hypertrophy of adipocytes in the modulation of metabolism through the stimulation of mitochondrial biogenesis and β-oxidation, which can lead to an induction of PPAR-α, and consequently, the control of differentiation and proliferation of adipose cells [47]. The VEI, SIN, OFEO, and OFAE groups also showed a significance of *p* < 0.001 when compared with GSC. The increase in body weight observed in the GSC group is related to the high daily calorie consumption of these animals in a diet composed of SFA, which contributes to the increase in adipose tissue [48].

Regarding the investigated biochemical markers of liver function, aspartate aminotransferase (AST), alanine aminotransferase (ALT), and kidney function (urea and creatinine), it was possible to observe that supplementation with a GSC-induced diet caused a significant increase in the levels of these biochemical markers, both hepatic and renal (Table 6). However, the treatments positively influenced these results, especially in the CHR and OFBO groups, demonstrating statistically significant results.

The increase in AST and ALT levels may be due to the administration of saturated fat associated with increased serum blood cholesterol levels, as well as the accumulation of fat in the liver. The SFAs generate suppression of the activity of hepatic LDL receptors in messenger RNA (rLDL), blocking intracellular and extracellular cholesterol transport since, as a result of cholesterol reduction, cells via SREBP-2 exhibit increased expression of genes involved in cholesterol synthesis [49,50].

### 2.3. Analysis of Lipid Profile and Atherogenic Index

The lipid profile of the SIN, CHR, OFEO, OFAE, and OFBO treatment groups showed significant results, both with *** *p* < 0.001, when compared to the GSC group, which showed high levels of TC, LDL-c, and TG (Figure 5). Coconut oil is a rich source of saturated fatty acids and its excess administration triggers cholesterol biosynthesis in the liver, causing hypercholesterolemia [51].

In a study conducted in Wistar rats submitted to treatment with coconut oil, high levels of CT and TG, as well as an increase in LDL-c and a decrease in HDL-c, were observed [51]. According to the study [52], GSC has a satisfactory effect in the induction of dyslipidemia as it has 92.64% saturated fat, which may increase the rate of cholesterol production as well as triglyceride levels; thus, it is a well-described model for inducing hyperlipidemia by GSC in animals.

In the present study, the data demonstrate that the CHR, OFEO, OFAE, and OFBO groups act to prevent GSC-induced hyperlipidemia. This effect may be associated with the presence of unsaturated, monounsaturated (MUFAs), and polyunsaturated (PUFAs) fatty acids in the composition of the tested oils, as in the granules, it is known that these fatty acids act in the modulation of lipid metabolism at levels of transcription of receptors such as PPARα, present mainly in the liver, essential in the capture and transport of fatty acids and β-oxidation, inducing levels of lipoprotein lipase (LPL), which has the function of hydrolyzing lipoprotein triglycerides into circulating fatty acids [53].

In addition to the AGIs providing great benefits in the lipid profile, the saturated fatty acid stearic (18:0), present in the chemical composition of the species *A. esculentus*, *E. oleracea*, *B. orellana*, does not promote hypercholesterolemia since the dehydrogenation of this fatty acid is faster than chain elongation, causing it to be converted to oleic acid (monounsaturated) in the liver, thus promoting the previous peroxisomal β-oxidation [54].

Another possible response pathway in metabolic homeostasis is through the antioxidant activity that is indirectly attributed to polyunsaturated fatty acids, which are present in the chemical composition of the studied species. These fatty acids act on vascular endothelial cells, decreasing inflammation and, in turn, the risk of atherosclerosis and cardiovascular diseases [55]. They act by preventing lipid peroxidation and eliminating reactive oxygen species (ROS) and reactive nitrogen species (RNS), which are responsible for the oxidation of LDL, a lipoprotein attributed to the formation of foam cells and consequently to the development of atherosclerosis [55,56]. ROS include superoxide anion, hydroxyl radical, and hydrogen peroxide. RNS are mainly nitric oxide and peroxynitrite [57,58].

In a study carried out on polyphenols present in *E. oleracea* oil, agents capable of neutralizing reactive oxygen species were identified, thus presenting antioxidant activity [59]. When obtaining the oil from the seeds of *A. esculentus*, it demonstrated antioxidant activity by the DPPH radical scavenging activity test. The tocotrienols present in the *B. orellana* species also showed antioxidant properties, improving oxidative stress in metabolic disorders and protecting cellular functions [60,61], contributing to the decrease in antiproliferative effects, immunoprotection, reducing the risk of cancer, and especially cholesterol [62,63]. Tocotrienols are present in CHR and OFBO.

Simvastatin was chosen as the standard drug; it acts as a widely used hypocholesterolemic agent. It is an inactive lactone that is converted into its corresponding β,δ-hydroxy acid in its active form, acting on the pathway of hepatic metabolism by cytochrome P450 after oral administration [64].

Simvastatin, known as statin, is a class of more effective drugs for the treatment of lipid alterations; it is a potent inhibitor of the enzyme 3-hydroxy-3-methyl-glutaryl-coenzyme A (HMGCoA) reductase, preventing the formation of mevalonate, which leads to a reduction in the hepatic synthesis of cholesterol, consequently acting in the increase in LDL-c receptors in hepatocytes, thus increasing its uptake from the circulation to restore cholesterol [65,66].

Regarding HDL-c, the results of this study showed a significant increase in the CHR, OFEO, OFAE, and OFBO groups when compared with the GSC groups. The CHR group had the highest levels of HDL-c (34.87 ± 3.4), followed by the OFBO (32.13 ± 4.36), OFAE (31.71 ± 4.15), and OFEO (31.29 ± 3.15 groups), thus indicating that the treatments were able to prevent the increase in LDL-c in this experimental model. These results are possibly related to the lipid composition of the species under study, which present in their composition high levels of monounsaturated fatty acids, such as oleic acid and eicosenoic acid (ω-9), and polyunsaturated fatty acids, such as α-linolenic acid (ω-3) and linoleic (ω-6), which can act in the treatment of cardiovascular diseases as well as dyslipidemia [30,67,68].

These unsaturated fatty acids are natural ligands of PPARα activation and can increase the transport of fatty acids, increase the expression of lipoprotein lipase receptors (LPLr), and suppress the levels of SREBP-1c responsible for the regulation of enzymes involved in the synthesis of fatty acids, in addition to increasing the transcription of the main apolipoproteins, such as ApoAI and ApoAII, present in HDL. Unsaturated fatty acids can also inhibit HNF-4α, resulting in reduced expression of genes involved in cholesterol biosynthesis in addition to increasing the transcription of major apolipoproteins, such as ApoAI and ApoAII, present in HDL [69].

As for the glycemic levels, we can observe significant satisfactory results; the lowest glycemia values were presented in the SIN, CHR, OFEO, OFAE, and OFBO groups, both with *p* < 0.001 when compared to the GSC (Table 7). The increase in glycemia in the GSC group may be related to the ingestion of SFAs, corroborating the accumulation of body fat, since the excess consumption of carbohydrates and lipids contributes to resistance to insulin action, consequently leading to a chronic and gradual increase in glycemia [70].

In addition to its role in lipid homeostasis, PPARα may influence glucose homeostasis, acting by directly regulating gluconeogenesis via stimulation of pyruvate dehydrogenase kinase 4 (PDK4) expression, which favors the use of pyruvate for gluconeogenesis to the detriment of fatty acid synthesis [71]. In animal models of insulin resistance, PPARα agonists increased insulin sensitivity, consequently increasing fatty acid oxidation in the liver, skeletal muscle, and pancreas by decreasing endogenous glucose [72].

Glucose homeostasis can also be attributed to the direct action of PPARγ on insulin-stimulated glucose disposal. The activation of PPARγ can increase the expression and translocation to the cell surface of the glucose transporter 1 (GLUT1) and glucose transporter 4 (GLUT4), thereby increasing the glucose uptake into adipocytes and muscle cells, as well as reducing plasma glucose levels [73].

### 2.4. Formation of Atherogenesis

Elevated levels of LDL-c in the blood are recognized as one of the risk factors for cardiovascular disease and are a consequence of the atherosclerosis process [60]. Atherosclerosis is a chronic inflammatory disorder that occurs in response to vascular endothelial aggression caused by an increased concentration of serum LDL in the arteries. After entry of LDL-lipoprotein into the intima of arteries, they undergo oxidation in the pro-oxidant environment, favored by an increase in reactive oxygen species (ROS) that oxidize LDL-cholesterol into oxidized LDL (LDLox), thus triggering the process of inflammation and the recruitment of immune cells, especially monocytes, where they differentiate into macrophages and phagocytose the LDLox, presenting lipids in their interior, and are now called foam cells, thus forming the atherosclerotic plaque [52,74,75].

The consumption of large amounts of saturated fat triggers the atherogenic process, especially in the abdominal aorta region, which is prone to plaque formation. The thoracic and abdominal arteries are the most affected during the process of atherosclerotic plaque formation [52,76].

The data presented in this study demonstrate compliance with the formation of atherosclerotic plaques in the vascular endothelium as well as the Plasma Atherogenic Index (AI) presented by the GSC group (1.30 ± 0.07) (Figure 6 and Table 7). However, when evaluating aortic atherogenesis, it was possible to demonstrate the great antiatherogenic potential of the CHR, OFEO, OFAE, and OFBO treatment groups due to the absence of plaque formation in the vascular endothelium, corroborating the data obtained from the AI of the treated groups.

This action can be explained by the capacity of the treatment groups (CHR, OFEO, OFAE, and OFBO) to present unsaturated fatty acids in their chemical composition, especially of the ω-3 and ω-6 series, as natural ligands of gene targets, which act in biological processes mainly related to lipid metabolism and the inflammatory process, thus playing a key role in several cardiovascular diseases. Contributing effectively to the preservation of endothelial function, reduction in plasma cholesterol, reduction in LDL-c, which is an important lipoprotein in the process of early atherogenesis, and stability of the atheroma plaque [77].

The SIN group did not form atherosclerotic plaque. This response can be explained as simvastatin acts to decrease the levels of LDLox and macrophages, actions that stabilize the atheromatous plaque, in addition to inhibiting low molecular weight G proteins of the Ras superfamily (Ras and Rho), key proteins involved in cell proliferation, differentiation, apoptosis, migration, contraction, and regulation of gene transcription, therefore improving vascular function. Simvastatins reduce the activity of NF-κB in inflammatory and vascular cells, as well as the inflammatory biomarkers C-reactive protein (CRP); however, they increase the endothelial expression of NO synthase (eNOS), increasing blood flow in the vessels, therefore improving hemostasis and recovery of endothelium-dependent vasoreactivity [78].

Lecithin, the surfactant used, is described as a complex mixture derived from crude soybean oil, consisting mainly of phosphatidylcholine, phosphatidylethanolamine, phosphatidylserine, and phosphatidylinositol, including other lipophilic substances such as glycolipids, triglycerides, or fatty acids, in addition to a hydrophilic fraction of phosphoric acid, glycerol, choline, and inositol [79,80]. Soy lecithin is used as a dispersant, emulsifier, stabilizer, marketed in different solid pharmaceutical forms, such as capsules, tablets, and granules, in the category of dietary supplements [81].

Due to their high lipophilicity, lecithins promote the physical coating of particles, especially those containing fat in their composition, in such a way that the oil, when in an aqueous medium, promotes a reduction in the surface tension between the solid and liquid phases, making these phases mix and form only one, having, therefore, important properties in the drug delivery system [82].

According to the literature, lecithins form fat-transport lipoproteins, allowing to reduce blood cholesterol levels. The phospholipids present play an important role during the intestinal absorption of lipids, thus facilitating the formation of micelles, increasing absorption, and having an activating action on the circulation, thus reducing the risk of cardiovascular diseases due to their emulsifying action, which does not allow the deposit of lipids to occur. Fat in blood vessels [83,84].

The lecithin group (VEI) used In this study was able to reduce the lipid profiles of TC, LDL, and TG, and increase HDL, showing significant values when compared to the GSC, but when compared with the treatments (CHR, OFEO, OFAE, and OFBO), it was not possible to obtain significant results. In addition to observing increased AI values as well as the presence of atheroma plaque when analyzed in Scanning Electron Microscopy (SEM), presenting, therefore, adverse results in the literature. Thus, we can infer that although lecithin presents responses in lipid levels, it did not demonstrate action in the reduction in atheromatous plaque; however, the treatment groups were more effective in modulating lipid metabolism as well as antiatherogenic activity obtained from the species under study; this response may be related to the attributed bioavailability, especially to the studied granules (CHR).

AGIs can modulate the inflammatory response by several mechanisms, such as via PPARα, which acts by inhibiting the expression of monocyte chemoattractant protein-1 (MCP-1) induced by the C-reactive protein (CRP) inhibition of endothelin 1 (ET-1) expression, inhibition of interleukin-6 (IL-6) release induced by interleukin-1 (IL-1), and inhibition of expression of vascular cell adhesion molecule 1 (VCAM-1) induced by liposaccharides (LPS) [85,86,87].

PPARα and γ can also attenuate the levels of interferon-gamma (IFN-γ) released by T lymphocytes; it also stimulates the expression of the cholesterol efflux regulator protein ABCA1 in foam cells in a liver X receptor (LXR)-dependent manner, promoting ApoA-I-mediated cholesterol efflux; regulates class B type 1 scavenger receptors (SRBI), which plays a role in esterified HDL uptake by the liver and cholesterol efflux from macrophages; as well as induces the secretion of lipoprotein lipase (LPL) and decreases the uptake of glycated LDL by macrophages [87,88,89].

In turn, the PPARγ in macrophages increases the expression of the scavenger receptor CD36/FAT, responsible for the uptake of LDLox and of fundamental importance in the differentiation of macrophages with characteristics of foam cells [90]. Later stages of atherosclerosis are also regulated by PPARα, inhibiting the activation of smooth muscle cell (SMC) proliferation, which is a key event in the development of atherosclerosis and its complications such as tissue factor, an important procoagulant; hence, PPARα may block atherothrombosis [91,92].

In addition to the AGIs promoting activity in the inflammatory response, compounds such as tocotrienols are important inflammatory modulators, as they are negative regulators of PPARγ receptors and can act by suppressing the formation of nitric oxide inducible by an inflammatory signal in macrophages, tumor necrosis factor alpha (TNF-α), as well as inhibition of nuclear factor-κB (NF-κB) activation, thereby stopping tissue inflammation [34,93,94]. In addition to these compounds, geranylgeraniol, present in the chemical composition of CHR and OFBO, can also promote the modulation of PPARγ receptors, act as an agonist, exert negative regulation via HMG-Coa-reductase, and activate enzymes involved in cholesterol biosynthesis, playing a key role in the development of atherosclerosis [34,35,95].

Inflammation is recognized as a key regulatory process in the face of several risk factors for atherosclerosis. C-reactive protein (CRP) is an inflammatory marker generated from the liver; it is an acute-phase reactant protein and responds immediately after the increase in nonspecific inflammation in the blood vessel, which may reflect the process of atherosclerotic development, therefore being recommended as a cardiovascular risk marker [36,37,96]. In view of the results obtained in this study, we can observe increased CRP values in the GSC group (0.84 ± 0.06), which corroborates the results by SEM in the appearance of atheromatous plaque, thus suggesting chronic inflammation in the aortic artery by atherogenic processes. However, we can observe decreased values of the CRP biomarker in the treatment groups CHR (0.41 ± 0.04), OFEO (0.49 ± 0.03), OFAE (0.50 ± 0.03), and OFBO (0.47 ± 0.04), as well as in the positive control group SIN (0.39 ± 0.04), demonstrating, therefore, a reduction in the inflammatory process and, consequently, antiatherogenic activity, as observed in the images obtained from the aortas by SEM.

### 2.5. Histopathological Analyses

In the histopathological analyses, in this study, the hepatic lobules, portal spaces, and well-delineated hepatic veins were easily visualized in the liver, and the hepatocytes formed confluent cords for the central-lobular vein.

The histopathological evaluation showed intense sinusoidal vascular congestion in animals from the GSC and SIN groups (Table 8). This alteration is characterized by the dilation of the hepatic sinusoids [38,39,97].

The occurrence of hepatic steatosis was also observed in the GSC and SIN groups. Hepatic steatosis consists of the accumulation of lipids in the cytoplasm of hepatocytes (Figure 7) [98]. It is the most frequent hepatic metabolic disorder and results in an imbalance between the synthesis of triacylglycerols from fatty acids and their low secretion by the hepatocyte in the form of lipoproteins. Triacylglycerols accumulate in hepatocytes by one or more of the following pathways: increasing the amount of circulating free fatty acids; reduced β-oxidation; or decreased synthesis or secretion of very low-density lipoproteins, which is primarily responsible for the secretion of triacylglycerols by the liver [99,100].

## 3. Materials and Methods

### 3.1. Obtaining Oils from Euterpe oleracea Martius, Abelmoschus esculentus L. Moench, Bixa orellana Linné, and Chronic SM^®^

The oils of *Euterpe oleracea* Martius (OFEO), *Abelmoschus esculentus* L. Moench (OFAE), *Bixa orellana* Linné (OFBO), and granulated (Chronic SM^®^) were provided by Ages Bioactive Compounds Co. (São Paulo, SP, Brazil). The extraction method has been standardized by the company and is under patent protection.

### 3.2. Obtaining Cocos nucifera L. Saturated Fat (GSC) and Simvastatin

The GSC was obtained from Cocos Empire Company-Mercado Municipal de Belo Horizonte, Minas Gerais, Brazil. The GSC extraction method was performed by mechanical pressing of the endocarp, followed by the addition of water 1:1 (*v*/*v*) and fat separation by heating at 80 °C [101]. Simvastatin 20 mg (SIN) (EMS Laboratory, Hortolândia/SP-Brazil) was purchased from a drugstore.

### 3.3. Transesterification of Oils from Euterpe oleracea Martius; Abelmoschus esculentus L. Moench by Enzymatic Catalysis

In a 25 mL vial, 500 mg of oils, 1.5 mL of ethanol, and 50 mg of *Candida antarctica* lipase (CAL-B ≥ 5000 U/g) were added. The reaction was maintained at 32 °C under magnetic stirring for 24 h. Subsequently, it was transferred to a separatory funnel, and the lower phase (glycerol) was separated, while the upper phase (mono-ester) was washed with distilled water (2 × 5 mL). The organic phase was dried with anhydrous sodium sulphate and filtered. Finally, the product was purified by flash column chromatography on silica gel with a mixture of ethyl acetate and n-hexane (9:1) as eluent [102].

### 3.4. Chemical Characterization of Euterpe oleracea Martius Oils; Abelmoschus esculentus L. Moench by Gas Chromatography

Analyses by gas chromatography coupled to mass spectrometry (GC-MS) were performed in a Shimadzu/GC 2010 instrument coupled to a Shimadzu/AOC-5000 autoinjector and a mass detector (Shimadzu MS2010 Plus) with electron impact (70 eV), equipped with a DB-5MS fused silica column (Agilent J&W Advanced 30 m × 0.25 mm × 0.25 µm) (65 kPa). The parameters were as follows: split ratio 1:20, helium as carrier gas, injection volume of 1.0 μL, injector temperature of 250 °C, detector temperature at 250 °C, initial column temperature of 120 °C, remaining for 2 min, and a heating rate of 5 °C/min up to 270 °C. The total analysis time was 36 min. The identification of fatty acid esters was determined by comparing the fragmentation spectrum with those contained in the GC-MS library (MS database, NIST 5.0) [103].

### 3.5. Quantitative Analysis of Total Tocotrienols Based on δ-Tocotrienol in Bixa orellana Linné Oil Samples and Granules (Chronic SM^®^)

For the quantitative analysis of the marker of interest, the samples were prepared with 2 solutions of the oil in hexane (Sigma-Aldrich, Lot. MKCF5755, GC grade, Darmstadt, Germany) at a concentration of 2 mg/mL and a solution of 1 mg/mL in 10 mL volumetric flasks. Then, 1 mL aliquots of the solution were transferred to each vial, and then 50 µL of BSTFA (Sigma-Aldrich, Lot. LC06356V, GC grade, Darmstadt, Germany), the derivatizing agent used to increase the volatility of the molecules, was added.

After the derivatization process, the samples were injected into a gas chromatograph coupled to a mass spectrometer (Shimadzu GC/MS–PQ2010SE) equipped with an RTX-5MS fused silica column. To estimate the tocotrienol content, calibration curves were prepared based on different concentrations that could be approximated by linear regression of the actual concentration of each sample.

The calibration curve was prepared from a 10 mg/mL stock solution of a standard containing 90% delta-tocotrienol and 10% gamma-tocotrienol in hexane. The chosen concentrations were 2.5 mg/mL, 1.0 mg/mL, 0.5 mg/mL, 0.25 mg/mL, and 0.1 mg/mL. All injected standards were derivatized with BSTFA to increase the volatility of the molecules and amplify detection by the equipment.

### 3.6. Quantitative Analysis of Geranylgeraniol in Samples of Bixa orellana Linné Oil and Granules (Chronic SM^®^)

For the quantitative analysis of the markers of interest, the samples were prepared in 2 solutions: a 1 mg/mL solution of the granulate (Chronic SM^®^) and a solution containing 0.5 mg/mL of the standardized oil, both in dichloromethane (Sigma-Aldrich, Lot. MKCF5755, GC grade, Germany) in 10 mL volumetric flasks. Then, 1 mL aliquots of each solution were filtered and transferred to each vial.

The samples were injected into a gas chromatograph coupled to a mass spectrometer (Shimadzu GC/MS–PQ2010SE) equipped with an RTX-5MS fused silica column. To quantify geranylgeraniol, a calibration curve was prepared based on different concentrations that could be approximated by linear regression of the actual concentration of each sample.

The calibration curve was prepared from a stock solution of 5 mg/mL of a standard of geranylgeraniol in dichloromethane. The chosen concentrations were 4 mg/mL, 2.0 mg/mL, 1.0 mg/mL, 0.5 mg/mL, 0.25 mg/mL, and 0.1 mg/mL.

### 3.7. Animals and Ethical Aspects

This study was approved by the Animal Use Ethics Committee of the Federal University of Amapá CEUA/UNIFAP under Protocol nº 03/2021 on 2 June 2021. The animals used were male Wistar rats (Rattus norvegicus albinus) from the Animal Investigation Multidisciplinary Center (CEMIB) at the University of Campinas, UNICAMP. Rats were kept in polyethylene cages placed in a ventilated, temperature-controlled cabinet (25 ± 2 °C) on a light/dark cycle (12/12 h), with free access to standard rodent food and water ad libitum.

### 3.8. Treatments and Induction of Dyslipidemia

Hyperlipidemia was induced by 2 mL of (GCS) daily for 40 days in all treatment groups. The animals were randomly divided into seven groups (n = 8 per group) and treated orally for 40 days: (1) treated with 0.5 mL of distilled water and 2 mL/day (GSC); (2) treated with vehicle 5% lecithin (VEI); (3) treated with simvastatin 20 mg/kg (SIN); (4) treated with 200 mg/kg of granules (CHR); (5) treated with 200 mg/kg (OFEO); (6) treated with 200 mg/kg (OFAE); (7) treated with 200 mg/kg (OFBO). The study carried out was based on the doses of previous works, thus substantiating the applied methodology [104].

### 3.9. Biochemical Analyses

The animals’ body weight and feed intake were measured every day throughout the treatment to calculate the weekly average. At the end of the experiment for the biochemical analyses, on the 41st day, the animals were fasted for 12 h and anesthetized with a combination of Ketamine and Xylazine (Cristália-Chemicals and Pharmaceuticals Ltda, Itapira, SP, Brazil) at a dose of 60 and 100 mg/kg, respectively, intraperitoneally. Blood samples (1.5 mL) were collected from the ocular plexus and centrifuged for 10 min (5000 rpm) for analysis of aspartate aminotransferase (AST), alanine aminotransferase (ALT), total cholesterol (TC) and fractions (LDL and HDL), triglycerides (TG), urea, glucose, and creatinine. The atherogenic index (AI) was calculated: Log (Triglycerides/HDL-Cholesterol) [105]. All tests were performed using LabTest kits and automated biochemical analysis equipment, model BS 380 (Mindray Biomedicina Electronics Co., Ltd., Nanshan, Shenzhen, China).

### 3.10. Analysis by Scanning Electron Microscopy (SEM)

Aortic isolation was performed from the aortic arch to the iliac bifurcation. The thoracic region was divided into 0.5 cm sections for analysis using equipment (SEM, Hitachi Model-TM3030PLUS) to detect atherogenic processes [56].

### 3.11. Histopathological Analyses

For histopathological evaluation, the liver was fixed in a 10% buffered formalin solution for 48 h and subsequently dehydrated in an alcohol battery, cleared in xylene, and embedded in paraffin (in lentils, Inlab brand) according to the methodology described by [106].

Afterwards, the blocks were cut in a rotary microtome (Slee Medical CUT 5062, Nieder-Olm, Germany) with disposable razors (Slee) at a thickness of 4 and 5 µm. The cuts were arranged on glass slides for optical microscopy, which were left for 25 min in an oven at 60 °C for drying and better adhesion of the material that was stained with HE (Harris hematoxylin-LABORCLIN and yellowish eosin-INLAB). The slides were observed in optical microscopes Olympus Micronal BX41 and photographed with an MDCE-5C USB 2.0 camera (digital).

Hepatic alterations were evaluated and quantified based on the following histological characteristics: sinusoidal vascular congestion, necrosis, and hepatic steatosis. The intensity of the histopathological characteristics was evaluated, in which the occurrences of alterations were expressed in crosses (0 to 3+), obtained through the average of three fields of random microscopes, being evaluated at a magnification of 200 times, considering the following graduation [107]:0+: no changes;1+: slight intensity changes (less than 25% of the analyzed field);2+: moderate intensity changes (25 to 50% of the analyzed field);3+: severe intensity changes (more than 50% of the analyzed field).

The data obtained were expressed as mean ± standard deviation and statistically analyzed using the Kruskal–Wallis test, followed by the Dunn method, always considering a *p* < 0.05.

### 3.12. Statistical Analysis

The results of the dyslipidemia induction experiments were expressed as mean ± standard deviation. Groups were compared using Analysis of Variance (One-Way ANOVA) followed by Tukey’s multiple comparison post-test; *p* < 0.05 was considered statistically significant between groups. Statistical programs used were GraphPad Instat and Prism (version 7.0).

## 4. Conclusions

Based on the results, we report that the CHR, OFEO, OFAE, and OFBO treatment groups can significantly reduce blood glucose, total cholesterol, triglycerides, and LDL levels, as well as increase the ability to synthesize HDL, in GSC-induced dyslipidemia models. The antiatherogenic capacity of CHR, OFEO, OFAE, and OFBO was demonstrated by the reduction in C-reactive protein, an inflammatory marker, and the absence of atheromatous plaque formation in the aorta. This was evaluated by scanning electron microscopy and the atherogenic index.

The antihyperlipidemic and antiatherogenic potential of the tested groups can be explained by pharmacological mechanisms already described, based on their main chemical compounds, such as the presence of unsaturated fatty acids, in addition to the presence of tocotrienols and geranylgeraniol. It is known that these substances are reported as natural ligands of important targets in the transcriptional regulation of metabolic pathways, therefore promoting antioxidant and anti-inflammatory activities and modulation of the lipid profile.

## 5. Patents

The extraction method and the standardization of the oils and granules are under patent protection confidentiality by the company Ages Bioactive Compounds Co. (São Paulo, SP, Brazil), with the number BR 102021001428-8.

## Figures and Tables

**Figure 1 molecules-28-06689-f001:**
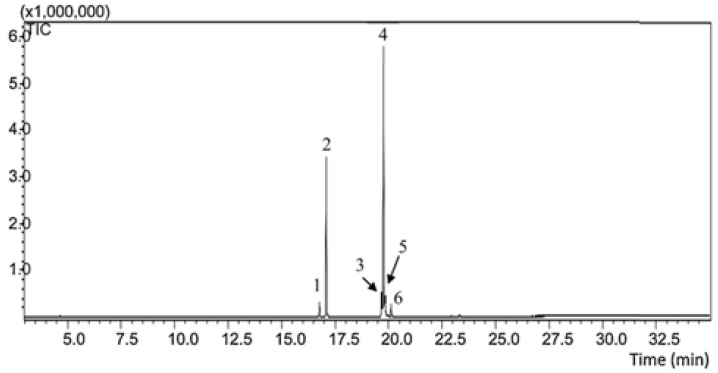
Analysis by gas chromatography coupled to mass spectrometry (GC-MS) of *Euterpe oleracea* oil. TIC = total ion chromatogram.

**Figure 2 molecules-28-06689-f002:**
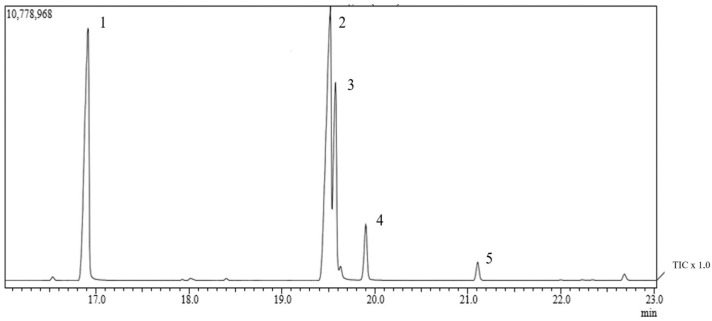
Analysis by gas chromatography coupled to mass spectrometry (GC-MS) of *Abelmoschus esculentus* oil. TIC = total ion chromatogram.

**Figure 3 molecules-28-06689-f003:**
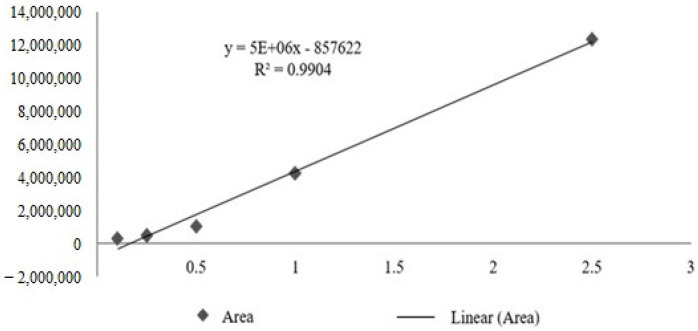
Calibration curve of total tocotrienols based on delta-tocotrienol.

**Figure 4 molecules-28-06689-f004:**
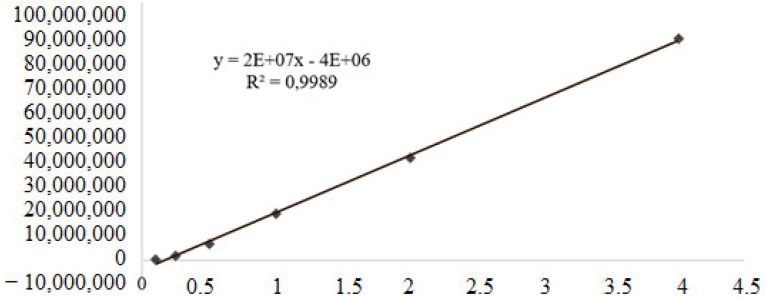
Calibration curve obtained for geranylgeraniol.

**Figure 5 molecules-28-06689-f005:**
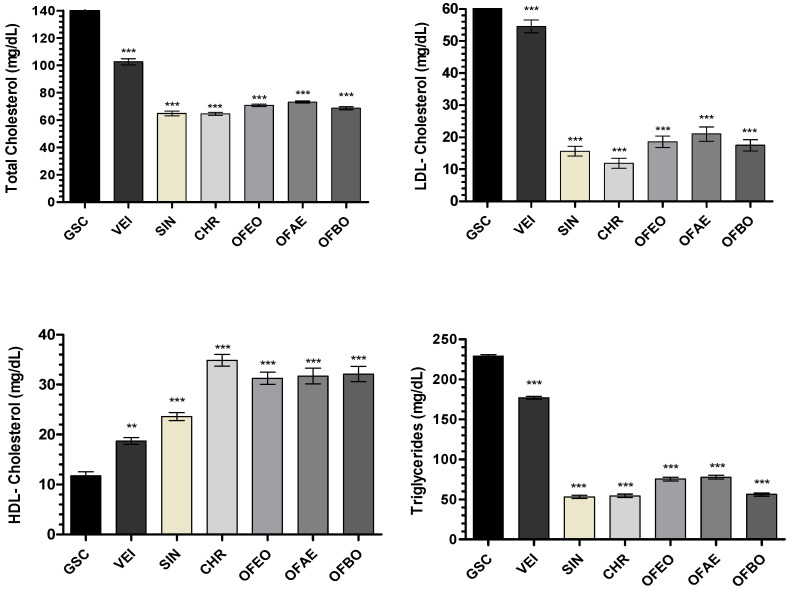
Effect of CHR, OFEO, OFAE, and OFBO treatment on total cholesterol, LDL, HDL, and triglycerides in GSC-induced dyslipidemia. Results presented as mean ± standard deviation. *** *p* < 0.001 and ** *p* < 0.01 represent statistically significant results compared to the GSC group. One-Way ANOVA test followed by the Tukey Multiple Comparisons post-test.

**Figure 6 molecules-28-06689-f006:**
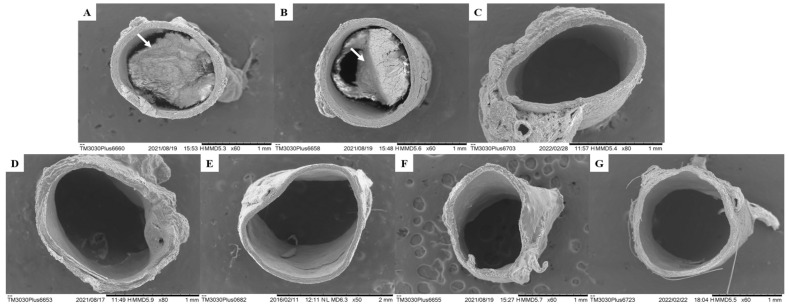
SEM images of cross-sections of the thoracic aorta (**A**) GCS, (**B**) VEI, (**C**) SIN, (**D**) CHR, (**E**) OFEO, (**F**) OFAE, and (**G**) OFBO. The white arrows indicate the formation of an atheroma plaque in the vascular endothelium.

**Figure 7 molecules-28-06689-f007:**
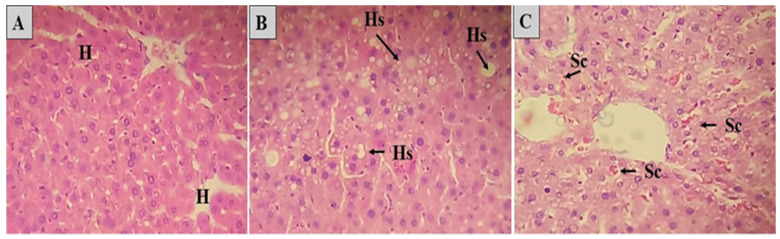
Histological section of the liver of animals submitted to the coconut oil-induced dyslipidemia test. In (**A**), normal liver, where hepatocytes are observed (H). In (**B**), altered liver, where Hs is observed: hepatic steatosis. In (**C**), altered liver, where Sc is observed: sinusoidal congestion. (H&E).

**Table 1 molecules-28-06689-t001:** Determination of FAEE equivalent to fatty acids (FAEE) in the composition of *Euterpe oleracea* oil analyzed by GC-MS.

Fatty Acid (n:i)	Peaks	Retention Time (min)	Concentration (%)
Palmitoleic (16:1)	1	16.75	2.62
Palmitic (16:0)	2	18.07	30.0
Linolenic (18:3)	3	19.68	5.9
Oleic (18:1)	4	19.78	54.32
Elaidic (18:1)	5	19.84	4.29
Stearic (18:0)	6	20.12	2.29
∑ Monounsaturated	-	-	61.27
∑ Polyunsaturated	-	-	5.9
∑ Unsaturated	-	-	67.83
∑ Saturated	-	-	32.83

aFAEE equivalent to the respective fatty acids and (n:i) number of carbon and unsaturation.

**Table 2 molecules-28-06689-t002:** Determination of FAEE equivalent to fatty acids (FAEE) in the composition of *Abelmoschus esculentus* oil analyzed by GC-MS.

Fatty Acid (n:i)	Peaks	Retention Time (min)	Concentration (%)
Palmitic (16:0)	1	19.91	33.55
Linoleic (18:2)	2	19.51	42.12
Oleic (18:1)	3	19.57	17.85
Stearic (18:0)	4	19.90	4.96
Eicosenoic (20:1)	5	21.10	1.53
∑ Monounsaturated	-	-	19.38
∑ Polyunsaturated	-	-	42.12
∑ Unsaturated	-	-	61.5
∑ Saturated	-	-	40.04

aFAEE equivalent to the respective fatty acids and (n:i) number of carbon and unsaturation.

**Table 3 molecules-28-06689-t003:** Average concentration and average content of total tocotrienols in OFBO and CHR.

Sample	Concentration (mg/mL)	Content (%)
OFBO	0.617448233	61.74482
CHR	0.46756	46.75485

**Table 4 molecules-28-06689-t004:** Average concentration and average content of geranylgeraniol of OFBO and CHR.

Sample	Concentration (mg/mL)	Content (%)
OFBO	0.872136	87.2136
CHR	0.1416946	14.1694

**Table 5 molecules-28-06689-t005:** Effect of CHR, OFEO, OFAE, and OFBO treatment on clinical parameters of Wistar rats with GSC-induced dyslipidemia.

Parameters	Water (mL)	Weight (g)	Food (g)
GSC	269.43 ± 13.26	375.02 ± 7.25	131.57 ± 6.66
VEI	252.36 ± 10.89	280.73 ± 7.20 ***	84.50 ± 1.55
SIN	251.60 ± 8.77 *	269.68 ± 4.03 ***	83.40 ± 1.92
CHR	245.21 ± 7.03 **	259.57 ± 4.00 ***	71.33 ± 2.86 ***
OFEO	237.33 ± 8.25 ***	265.41 ± 5.11 ***	83.67 ± 1.97
OFAE	238.90 ± 8.10 ***	270.39 ± 4.62 ***	96.14 ± 4.22
OFBO	249.19 ± 5.93 *	243.52 ± 4.25 ***	76.60 ± 1.14 ***

Results presented as mean ± standard deviation. *** *p* < 0.001, ** *p* < 0.01, and * *p* < 0.05 represent statistically significant results compared to the GSC group. One-Way ANOVA test followed by the Tukey Multiple Comparisons post-test.

**Table 6 molecules-28-06689-t006:** Effect of treatment with CHR, OFEO, OFAE, and OFBO on biochemical parameters of Wistar rats with GSC-induced dyslipidemia.

Parameters	Urea (mg/dL)	Creatinine (mg/dL)	AST (U/dL)	ALT (U/dL)
GSC	38.29 ± 5.96	0.90 ± 0.04	141.08 ± 7.27	59.11 ± 7.37
VEI	33.29 ± 2.29	0.82 ± 0.05 *	136.50 ± 8.64	51.97 ± 4.20
SIN	27.75 ± 5.26 **	0.80 ± 0.04 ***	128.70 ± 8.58 *	46.11 ± 5.65 ***
CHR	28.63 ± 5.68 *	0.85 ± 0.03 **	131.60 ± 5.12	48.64 ± 3.00 **
OFEO	32.00 ± 4.16	0.81 ± 0.04 **	132.55 ± 4.25	51.44 ± 3.56
OFAE	30.71 ± 5.19	0.83 ± 0.03 *	133.15 ± 6.67	53.34 ± 2.59
OFBO	29.13 ± 3.31 *	0.81 ± 0.05 ***	132.50 ± 5.48	50.49 ± 4.26 *

Results presented as mean ± standard deviation. *** *p* < 0.001, ** *p* < 0.01 and * *p* < 0.05 represent statistically significant results compared to the GSC group. One-Way ANOVA test followed by the Tukey Multiple Comparisons post-test.

**Table 7 molecules-28-06689-t007:** Effect of treatment with CHR, OFEO, OFAE, and OFBO on biochemical parameters and Atherogenic Index in the plasma of Wistar rats with dyslipidemia induced by GSC.

Parameters	Glucose (mg/dL)	PCR (mg/dL)	Atherogenic Index of Plasma (AIP)
GSC	161.29 ± 9.23	0.84 ± 0.06	1.30 ± 0.07
VEI	143.00 ± 6.81 ***	0.62 ± 0.04 ***	0.98 ± 0.04 ***
SIN	129.25 ± 5.15 ***	0.39 ± 0.04 ***	0.23 ± 0.07 ***
CHR	114.88 ± 2.95 ***	0.41 ± 0.04 ***	0.19 ± 0.07 ***
OFEO	126.00 ± 3.37 ***	0.49 ± 0.03 ***	0.38 ± 0.03 ***
OFAE	122.86 ± 2.04 ***	0.50 ± 0.03 ***	0.39 ± 0.03 ***
OFBO	117.13 ± 4.05 ***	0.47 ± 0.04 ***	0.25 ± 0.08 ***

Results presented as mean ± standard deviation. *** *p* < 0.001 represent statistically significant results compared to the GSC group. One-Way ANOVA test followed by the Tukey Multiple Comparisons post-test.

**Table 8 molecules-28-06689-t008:** Histopathological study of the liver of animals submitted to the *Cocos nucifera* L. (GSC)-induced dyslipidemia test.

Group/Change	GSC	VEI	SIN	CHR	OFEO	OFAE	OFBO
Sinusoidal congestion	2 ^a^	0	1 ^a^	1	0	0	0
Steatosis	2 ^a^	1	2 ^a^	0	1	1	1
Necrosis	0	0	0	0	0	0	0

^a^ Represents a statistically significant difference *p* < 0.05 of the groups in relation to the negative control group (Kruskal–Wallis test, method).

## Data Availability

Not applicable.

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
