# Peer review of "Effect of the Association of Fixed Oils from Abelmoschus esculentus (L.) Moench, Euterpe oleracea Martius, Bixa orellana Linné and Chronic SM® on Atherogenic Dyslipidemia in Wistar Rats"

_molecules, 2023, doi:10.3390/molecules28186689_

Round 1

Reviewer 1 Report

Comments and Suggestions for Authors

This research work is interesting and the presented information is of a certain scientific and practical interest. The experimental methodology is well explained and seems adequate to the objectives. Results are clearly discussed and linked to previous studies, and the paper can be published after minor changes.

Specific:

- Line numbers should be added to the manuscript for ease of review.

- Page 3, 13, 16: Change “Nascimento et al. (2008)”, “Oda (2008)” and “Rhoden et al., (2000)” to number, References should be written in a uniform style, according to the journal's guidelines. Please check all manuscript. 

- Please write full name before abbreviation at first mention.

- Avoid, where is possible, to use abbreviations.

- Page 3: “Saturated fatty acids (SFA) are obtained in the diet,…………………………….., being classified into ω-3, ω-6 and ω-9 [30]. This paragraph should be deleted, because this aspect is well known.

Author Response

We appreciate the suggestions, and all comments have been taken into account.

Reviewer 2 Report

molecules-2539774-peer-review-v1

Article

Atherogenic dyslipidemia in Wistar rats: study of the association of fixed oils from Abelmoschus esculentus (L.) Moench., Euterpe  oleracea Martius., Bixa orellana Linné and Chronic SM®

The manuscript should be improved in some of its parts.

1-      Introduction.

After references 23 and 24, a paragraph should be included where it is mentioned or described what it is granules (Chronic SM®, what it is marketed for and other characteristics of the product, considering that it is an open article.

https://agesbioactive.com/chronic/ Chronic® is a natural product, developed from the phytocomplex oily extract of Bixa orellana L., a plant from the Amazon biome, a source of Geranylgeraniol and Delta-tocotrienol, molecules with excellent anti-inflammatory functionality, effective antioxidant power and impact on optimizing function. mitochondrial.

2-      The title should be revised and mention the relationship between Bixa orellana Linné and Chronic SM.

3-      Abstract

This study investigated the effects of treatments with oils from A. esculentus, E. oleracea, B. orellana and granules (Chronic SM®) on saturated fat[1]induced dyslipidemia Cocos nucifera L.

This paragraph could be revised and improved in its wording.

4-      Assay: 3.8 Treatments and Induction of Dyslipidemia

The trial is a single dose study, with no dose-response studies.

The authors should justify the choice of the single dose evaluated. Any relevant bibliography that supports your choice could be mentioned in this section

Regarding simvastatin: Mention the trademark, company and country from which you acquired this patron drug

5-      3.12 Statistical analysis

The results of the dyslipidemia induction experiments were expressed as mean ± standard deviation. Groups were compared using Analysis of Variance (One-Way  ANOVA) followed by Tukey's multiple comparison post-test and p<0.05 was considered statistically significant between groups. Statistical programs used were GraphPad Instat and Prism (version 7.0).

The presentation of the p data should be reviewed( ˂; ˃ or =).

In this section it says p less than 0.05 (pË‚0.05) , however in the abstract and in the graphs it says p greater than 0.001 (***p>0.001 Table 7); (***p>0.001 and **p>0.01 Figure 5)  and others.

This should be unified, perhaps the exact value of p for each trial should be placed, and thus confusion would be avoided.

6-      Considering that granules Chronic SM® are marketed, is there a relationship between the dose tested and the doses consumed or suggested for consumption by the company that produces them?.

Include a short paragraph about this suggestion in the discussion.

Once the suggestions are considered, the work could be evaluated again for acceptance.

Moderate editing of English language required

Author Response

The corrections made are highlighted in the manuscript.

Reviewer 3 Report

Dear colleagues!

I have a number of questions and remarks on the merits.

1. On what basis were the diagnoses included in the group selected?

2. Big questions on the list of references and related sections "Relevance, Discussion". What was the study period? Unfortunately, there are works in the links that are more than 20 years old and I have doubts about their reliability for this review.

3. It would be appropriate to point out the epidemiology of Atherogenic dyslipidemia over time

4. The section "Materials and Methods" is not described, there is no null hypothesis and definition of the sample size.

Unfortunately, you need to revise your article.

Author Response

(The authors gave the same response as above.)

Round 2

Reviewer 2 Report

The paper should be accepted in its current state; suggested changes have been considered.

Minor editing of English language required

Reviewer 3 Report

Hello

Informative responses are satisfactory.

However, I draw the attention of the authors to the fact that although some beliefs are classical, modern research methods often refute them. And to argue that what is old correct, which must always be quoted, is not correct.

If you write about the history of medicine, then there will be fewer questions. Now out of 107 sources, 36 are older than 15 years (33%).